# Optofluidic Particle Manipulation Platform with Nanomembrane

**DOI:** 10.3390/mi13050721

**Published:** 2022-04-30

**Authors:** Zachary J. Walker, Tanner Wells, Ethan Belliston, Sage Romney, Seth B. Walker, Mohammad Julker Neyen Sampad, S M Saiduzzaman, Ravipa Losakul, Holger Schmidt, Aaron R. Hawkins

**Affiliations:** 1Department of Electrical and Computer Engineering, Brigham Young University, Provo, UT 84602, USA; tnwells@nwi.net (T.W.); ethanbelliston99@gmail.com (E.B.); sageromney@gmail.com (S.R.); walkerseth21@gmail.com (S.B.W.); hawkins@ee.byu.edu (A.R.H.); 2School of Engineering, University of California, Santa Cruz, CA 95064, USA; msampad@ucsc.edu (M.J.N.S.); smsaidzaman@gmail.com (S.M.S.); rlosakul@ucsc.edu (R.L.); hschmidt@soe.ucsc.edu (H.S.)

**Keywords:** optofluidic, microfluidic, nanomembrane, NEMS, lab-on-a-chip, biosensor

## Abstract

We demonstrate a method for fabricating and utilizing an optofluidic particle manipulator on a silicon chip that features a 300 nm thick silicon dioxide membrane as part of a microfluidic channel. The fabrication method is based on etching silicon channels and converting the walls to silicon dioxide through thermal oxidation. Channels are encapsulated by a sacrificial polymer which fills the length of the fluid channel by way of spontaneous capillary action. The sacrificial material is then used as a mold for the formation of a nanoscale, solid-state, silicon dioxide membrane. The hollow channel is primarily used for fluid and particle transport but is capable of transmitting light over short distances and utilizes radiation pressure for particle trapping applications. The optofluidic platform features solid-core ridge waveguides which can direct light on and off of the silicon chip and intersect liquid channels. Optical loss values are characterized for liquid and solid-core structures and at interfaces. Estimates are provided for the optical power needed to trap particles of various sizes.

## 1. Introduction

Optofluidic devices have been utilized for a variety of Lab-on-a-chip biological detection applications such as refractive index detection (RI) [1], surface-enhanced Raman spectroscopy (SERS) [2], and bulk fluorescence microscopy [3]. Optical methods have been developed to manipulate, sort and trap particles for improved detection schemes. Optical traps such as optical and optoelectronic tweezers have been explored to guide particles and hold them for further observation [4,5,6,7].

Many optofluidic platforms such as surface-micromachined anti-resonant reflecting optical waveguides (ARROW) and wafer-bonded ARROW waveguides have proven efficient at guiding light in liquid channels with low loss. ARROW waveguides with typical cross sections of 12 µm by 5 µm, use interference to contain light in liquid cores and have achieved waveguiding with losses in the ~2–5 cm^−1^ range. This method tends to have lower loss and allows for higher optical intensity waveguide transmission [8]. Another optofluidic channel fabrication method uses wafer bonding. In this method, channels are constructed by etching a trench into one substrate and bonding a second planar substrate to the first. Both substrates are often coated in ARROW layers prior to bonding but other techniques such as integrated Bragg, Teflon^TM^, and metal have all been used to construct effective liquid waveguides through wafer bonding [9]. Liquid waveguides utilizing Teflon^TM^ are capable of guiding light over 2 cm [10]. However, these platforms are not capable of easily hosting a thin top membrane with nanoscale thickness. Typical microfluidics incorporate buried channels or have thick membranes which proves problematic when trying to gain access to particles in the fluid. Thick silicon dioxide (~4 µm) is needed for strong channel walls, and this causes the membranes to be on the same order of thickness due to blanket silicon dioxide growth processes. Dry and wet etching can be rough and imprecise when trying to remove material of this thickness, even with a stop etch layer. Thin membranes often have low yields due to their inability to withstand the sacrificial material removal process. Wafer-bonded optofluidics have similar challenges as thick substrates must be thinned and etched in order to achieve nanomembrane results.

The device demonstrated in this paper introduces a particle manipulation platform with the desirable feature of an ultra-thin encapsulating membrane. Our platform is designed so that light from an optical fiber is coupled into an on-chip solid-core (SC) waveguide. The SC waveguide intersects and couples light into a microfluidic liquid-core (LC) channel. The LC has a cross section of 10 µm by 10 µm, the small size of which helps with trapping and holding particles by concentrating the radiation pressure. The LC channel acts as a leaky-mode waveguide structure and does not guide light efficiently. Due to the high optical loss, this channel is only suitable for transmitting light over short distance but can still be used in applications such as pushing, sorting, and trapping particles. Particles flowing through the liquid channel will experience optically induced radiation pressure from light introduced by the intersecting SC waveguide. Radiation pressure causes momentum to be transferred to matter inside the fluid. Though the force is relatively weak, the momentum is enough to guide micro/nanoscale particles and hold them against the channel wall for further interrogation [11]. In our design, this intersection region contains a protrusion cavity which provides a physical mechanism to isolate particles from the liquid flow and assist in trapping.

The optofluidic section of our new platform features a 300 nm, naturally forming, meniscus membrane, making it a unique optofluidic device. Building off a previous microfluidic design our group developed previously, a sacrificial polymer is introduced into the channels by utilizing the capillary effect [12]. The polymer forms a meniscus shape and is used as a template for membrane formation. Silicon dioxide is then grown over the channel using plasma enhanced chemical vapor deposition (PECVD), forming a membrane that matches the meniscus shape of the polymer template. When the polymer is removed, a 300 nm flexible silicon dioxide membrane is left suspended above the channel.

This platform represents an extreme version of a surface micromachining process requiring layers of only a few hundred nanometers. This is advantageous as it allows the liquid in the channel to be close to the exterior of the channel, making targets in the liquid easily accessible. Thin meniscus membrane implementation can be used in application such as integrating nanopores with fluid channels [13]. Thin membranes are advantageous in solid-state pore applications as smaller pores can be drilled with higher resolution for improved detection capabilities as shown in Figure 1. Other applications include top electrodes which can be added over the membranes for electrophoretic focusing [14,15] or electrochemical measurements [16], taking advantage of the close proximity between particles and electric fields. Filtration of particles can also be performed using a nanopore array in the membrane [17] or by utilizing the porous nature of the silicon dioxide itself [18]. Note that the net porosity of such films increases dramatically with thickness.

## 2. Nanofabrication Process

Figure 2 depicts the method used to fabricate the optofluidic particle manipulator. The microfluidic design is first patterned on a 100 mm silicon wafer by way of photolithography utilizing AZ Nlof 2020 photoresist. The features of note include 3 mm long microfluidic channels attached to inlet and outlet openings, reservoir walls, and an optofluidic region with a protrusion cavity on each end for holding particles. The channels are then anisotropically etched in an STS ICP Multiplex ASE RIE/ICP tool. Silicon side walls are formed to create the basis of the microfluidic channel (Figure 2a). The walls are 10 µm high and form a 14 µm wide channel. The wafer is then placed in a furnace where thermal oxidation occurs (Figure 2b).

Ensuring complete oxidation conversion of the SC to LC interface, as shown in Figure 3d,e, is critical so that light can be coupled efficiently into the optofluidic region. Silicon is opaque to visible electromagnetic radiation and will scatter incoming light, preventing transmission into the liquid channel. Thermal oxidation converts silicon to silicon dioxide in a furnace at 1100 °C. A wet oxidation process is used which allows steam to filter into the chamber where the chemical interaction of silicon with water results in thermal silicon dioxide on the substrate surface and produces hydrogen gas as a byproduct. The side wall is 2 µm thick at the interface. According to the BYU Cleanroom silicon dioxide growth calculator (link: https://cleanroom.byu.edu/oxidetimecalc accessed on 1 March 2022), this conversion should take approximately 10.5 h.

After this, 3 µm of PECVD silicon dioxide with a 1.51 refractive index is deposited (Figure 2c). Silicon dioxide refractive indices were calculated by use of an ellipsometer. This step is for the formation of the solid-core waveguides as shown in Figure 3. Next, droplets of SU8-2000.5, a photo polymer, are placed into the inlet and outlet reservoirs with a syringe. Utilizing the capillary effect, the polymer flows down the channel forming a meniscus shape that is used as a sacrificial template for membrane formation (Figure 2d). This nanomembrane technique was developed and demonstrated by our group previously [12]. The open channels extend nearly a millimeter into the reservoirs to prevent capillary flow on the outsides of the channels. Next, 300 nm of PECVD silicon dioxide is grown to form the meniscus membrane (Figure 2e).

The optofluidic region where the particle manipulation occurs requires a thin membrane to facilitate the applications discussed previously, but without reinforcement, the membrane is susceptible to cracks. A protection step is added to strengthen the channel membrane while maintaining the thin membrane in the critical intersection. Once the thin silicon dioxide membrane has been deposited, SU8 10 is patterned over the optofluidic region. Two micrometers of low index PECVD silicon dioxide is then grown over the entire channel. The 2 µm of low index silicon dioxide was subsequently dry etched in the optofluidic region, using the SU8 10 as a stop etch. Simultaneously, the inlet and outlet reservoir areas are etched, exposing the sacrificial SU8 2000.5 core. The wafer is then placed in a piranha solution, removing both SU8 sacrificial layers. This results in a suspended membrane that covers a hollow core liquid channel (Figure 2f). The optofluidic region hosts a 300 nm membrane while the rest of the microfluidic channel is protected by 2 µm of silicon dioxide.

The construction of the intersecting SC waveguides occurs simultaneously along the microfluidic fabrication process. The pedestal undergoes oxidation conversion, creating a low index silicon dioxide (*n* = 1.44) as cladding for the waveguide. High index PECVD silicon dioxide (*n* = 1.51) is deposited and used as the waveguide core (Figure 3a). The ridge waveguide design is then patterned and etched into the silicon dioxide, forming a rectangular cross section of 10 µm by 3 µm (Figure 3b). Low index (*n* = 1.44) PECVD silicon dioxide is added during the protection step and serves as a stronger membrane for the liquid channel and a cladding layer for the waveguide (Figure 3c). The pedestal on which the waveguides reside is etched at the same time as the channel wall. The waveguide pedestal is ~3 µm lower than the LC so the incoming light will enter below the meniscus membrane (Figure 3d). The SC to LC interface must be completely converted to silicon dioxide as shown in Figure 3e.

## 3. Optical Manipulation Region

The various fabrication steps demonstrated in the previous section come together and result in an optical manipulation platform. The functionality and design of the device is shown in Figure 3. Liquid flows from one reservoir to another by way of electrophoretic transport. Particles are introduced into the inlet reservoir where they are subsequently transferred through the channel. The bend in the channel at the optofluidic region causes the velocity of the particles to slow which makes for easier optical trapping. Faced with optical stimulation from the solid-core waveguides, the particles experience radiation pressure and are pushed into the protrusion region where they are held. The protrusion cavity provides a physical mechanism to help isolate the particles from the fluid stream. The protrusion has a depth of 20 µm, making the length of the optofluidic region ~30 µm total. The thin membrane encases the protrusion region outlined in yellow in Figure 4.

## 4. Optical System Loss

The ARROW optofluidic platform has an extensive history with a variety of optical sorting and trapping applications [8,9,19]. However, the fabrication of a nanoscale membrane using traditional ARROW methods has proved challenging. The optofluidic method discussed in this paper moves away from traditional ARROW designs to attain a thin membrane in the optofluidic region.

Both the LC channel and SC waveguide on the optofluidic particle manipulator were characterized to determine their optical loss coefficients. To do this, a 4 mm optofluidic channel was fabricated and tested as show in Figure 5. The locations of the various propagation losses and interface efficiencies are noted.

A typical method for characterizing ARROW devices is that of scattered light imaging developed previously [20,21]. A linear camera (FASTCAM S3) is used to capture an image of the SC waveguide and LC channel segments while they are guiding light and the LC is filled with water as shown in Figure 6. The image is processed and used to measure loss in each section of the waveguide. The optical intensity of each pixel along the SC and LC waveguides is extracted using MATLAB and then plotted. The optical loss can be ascertained by fitting the data to a line of best fit. Figure 7 demonstrates a typical LC channel intensity plot for a single optofluidic manipulation platform chip. The light coming through to the liquid channel quickly drops off due to the large optical loss, proving that this optofluidic particle manipulator is best suited for proximity applications.

The optical testing utilizes a 635 nm ThorLabs fiber coupled laser source (model S1FC635) at 0.75 mW attached to a single mode optical fiber. Individual devices are cleaved at the SC so that the fiber can be edge coupled to the waveguide on chip using three axis optical stages for alignment. The optical throughput can be measured as the light exits the chip, passes through an objective lens, and is focused onto a ThorLabs photodiode amplifier (model LMR1). Eight devices were tested and measured to determine the loss of both the SC waveguide and LC channel sections as shown in Figure 8. Because there is some variability in alignment and curve fitting image data points, an uncertainty value was calculated by testing the same device eight times. The resulting loss values were used to calculate the uncertainty. The average loss for the solid-core waveguides was 0.50 cm^−1^ with an uncertainty of 9.65% and a standard deviation of 0.14 cm^−1^. The average loss of the liquid-core channels was 6.61 cm^−1^ with an uncertainty of 11.49% and a standard deviation of 2.53 cm^−1^. Using these results, the solid to hollow core interface coefficient can be found using
(1)$$\kappa_{i}=\sqrt{T*e^{\alpha_{sc}l_{sc}}*e^{\alpha_{hc}l_{hc}}}$$
where *T* is the transmission, $\alpha_{sc}$ is the loss in the solid core, $l_{sc}$ is the length of the solid core, $\alpha_{hc}$ is the loss in the hollow core and $l_{hc}$ is the length of the hollow core. The average interface coefficient ($\kappa_{i}$) for these devices was 0.34. Typical 12 µm by 5 µm cross sectional ARROW loss values will be ~2 cm^−1^ in the LC and ~0.5 cm^−1^ in the SC with an interface coefficient of 0.56 [20]. The variability of the LC loss is most likely caused from deformities in the membrane and sidewalls which can be a source of significant loss.

One observation of note is that the loss should improve in the LC waveguide if the entire channel wall were to be converted to thermal silicon dioxide. Because the oxidation growth time is slow, this has been difficult to achieve on our current iteration, but if the silicon walls started with a reduced thickness, the entire wall could be converted to silicon dioxide and improve the loss coefficients to align with previous ARROW values more closely.

The optical power required to trap particles can be determined by the forces acting on the particle. The drag force acting on the particle originating from the liquid flow is given by Stokes’ law
(2)$$F_{Flow}=6\pi \mu rv$$
where µ is the viscosity of the liquid, r is the radius of the particle, and ν is the velocity of the liquid flow. The force from the SC waveguide radiation pressure acting on the particle is given by
(3)$$F_{Laser}=Q\pi \frac{r^{2}}{c}I$$
where *Q* is a dimensionless variable relating the momentum transfer efficiency, *n* is the index of refraction of the liquid, *c* is the speed of light, and *I* is the intensity of the laser [22].

The simulated zero-order E field intensity mode profiles for both the SC and LC are shown in Figure 9a,b using Ansys Lumerical FDE. The simulated optical force spatial distribution is also plotted using Equation (3) in Figure 9c, showing that the optical force loss is negligible over the 10 µm channel of our optofluidic region. The experimental SC and LC loss values obtained from top view scattering results were used to determine the optical intensity in equation 3 for this simulation.

To estimate the minimum power required to trap a particle, we focus on the worst-case scenario where a particle is on the edge of the channel opposite the protrusion. A particle reaches a terminal horizontal velocity when $F_{Flow}$ equals $F_{Laser}$. When the horizontal and vertical velocities are equal, we can ensure that particles are pushed into the trap region as shown in Figure 10.

Figure 11 shows the theoretical minimum trapping power over a range of particle radii for typical flow velocities based on Equations (2) and (3). Again, this is estimated by the worst-case scenario where a particle is on the edge of the channel opposite the protrusion. The simulation parameters are not based on experimental data. Smaller particles require more optical power to be pushed into the protrusion cavity and become trapped due to their limited surface area. The higher the liquid flow velocity, the more power is required to trap particles.

## 5. Conclusions

The optofluidic particle manipulator demonstrated in this paper offers exciting new opportunities for thin membrane optofluidics. The SC waveguide was able to perform well with an average loss value of 0.5 cm^−1^. Although the loss is high in the LC section of the device (6.61 cm^−1^), proximity application will have enough radiation pressure entering the protrusion cavity that trapping, sorting, and holding particles is possible. Increased power from the laser source can be used to compensate for the loss in the LC. The optical manipulation is selective and temporary giving it the ability to capture or release particles and concentrate them at a specific point. Finally, the optical power needed to trap particles of differing sizes was demonstrated. Particles of 0.5 µm diameter and 1 µm diameter would need ~0.5 (a.u.) and ~3 (a.u.) (log scale) of power respectively for minimum trapping at 1 cm/s flow velocity. The optical manipulation combined with the realization of a 300 nm membrane over the trapping section will be advantageous for near surface optofluidic technology.

## Figures and Tables

**Figure 1 micromachines-13-00721-f001:**
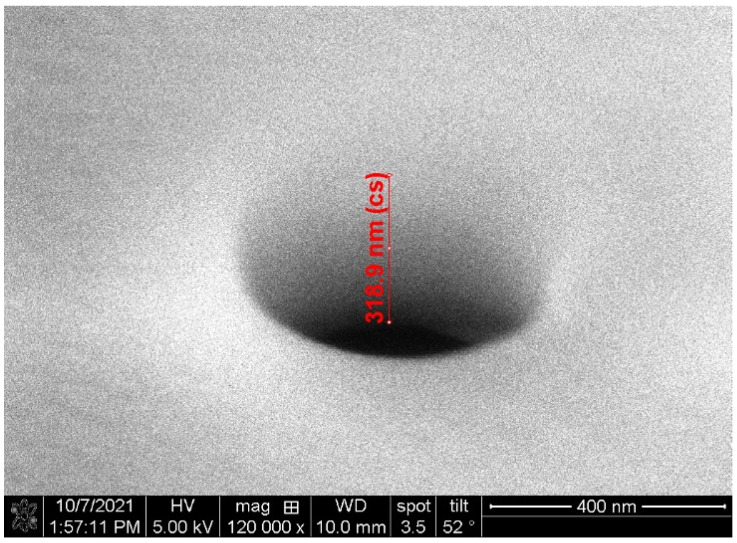
Scanning electron micrograph (SEM) of a solid-state pore drilled into a nanomembrane on an optical manipulation device using a focused ion beam (FIB).

**Figure 2 micromachines-13-00721-f002:**
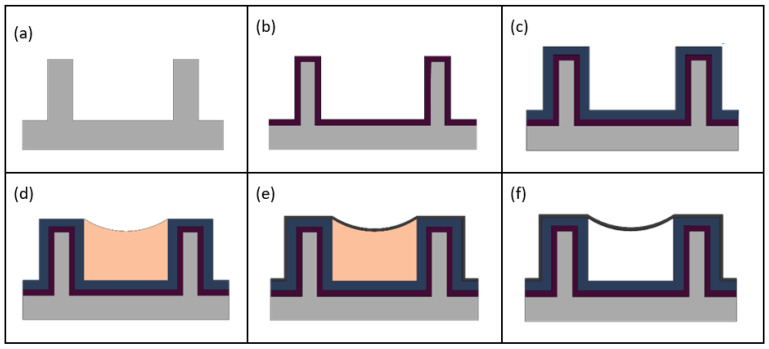
Fabrication flow diagram. (**a**) Microfluidic side walls are etched into silicon substrate. (**b**) Thermal oxidation converts silicon into silicon dioxide. (**c**) High index PECVD silicon dioxide is grown for SC waveguides as shown in Figure 2. (**d**) SU8 2000.5 is introduced into the channels by way of spontaneous capillary action. (**e**) A silicon dioxide membrane is formed by PECVD deposition. (**f**) SU8 2000.5 is sacrificially etched, leaving an intact membrane covering a hollow-core channel.

**Figure 3 micromachines-13-00721-f003:**
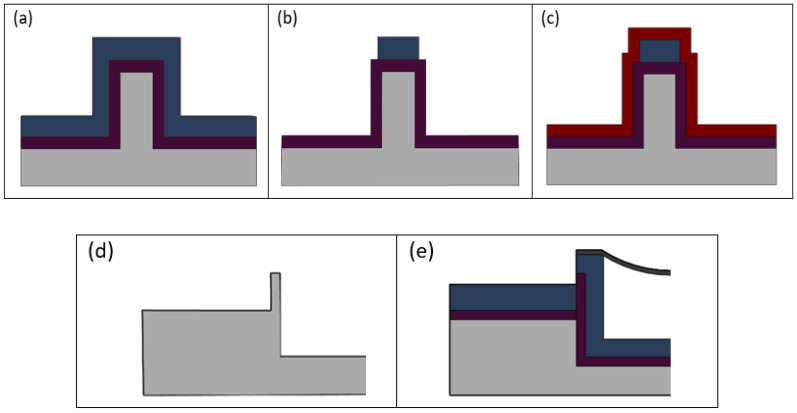
Solid-core ridge waveguide fabrication. (**a**) A pedestal is etched into silicon and then converted into silicon dioxide by way of thermal oxidation. High index PECVD silicon dioxide is grown over the thermal silicon dioxide. (**b**) High index silicon dioxide is patterned and etched to form the waveguide core. (**c**) Low index PECVD silicon dioxide is grown over the high index silicon dioxide as a cladding layer. (**d**) SC to LC intersection after etching in silicon and before thermal oxidation. Intersection side wall must be thin enough that complete thermal oxidation from silicon to silicon dioxide occurs. (**e**) Complete thermal oxidation of interface. Ridge waveguide can transmit light into the LC through interface.

**Figure 4 micromachines-13-00721-f004:**
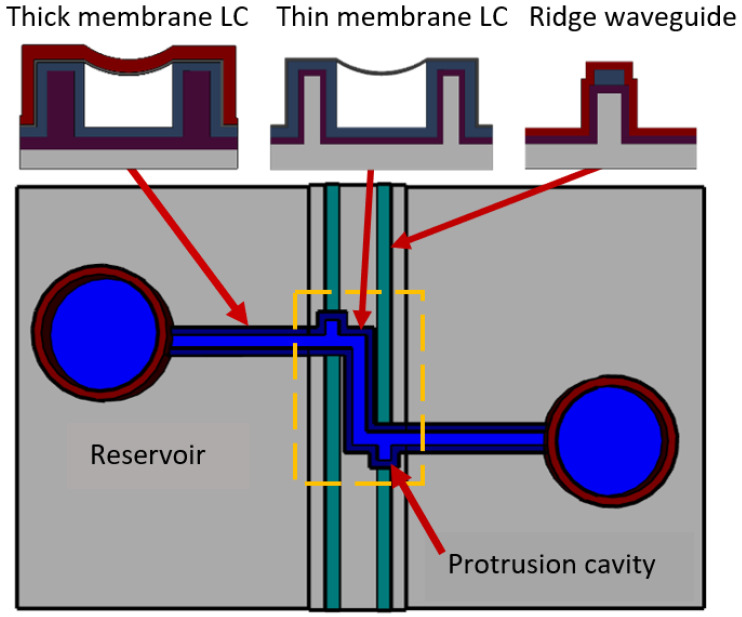
Optofluidic particle manipulation chip layout. Location of the thick membrane LC, thin membrane LC, and ridge waveguide all defined.

**Figure 5 micromachines-13-00721-f005:**
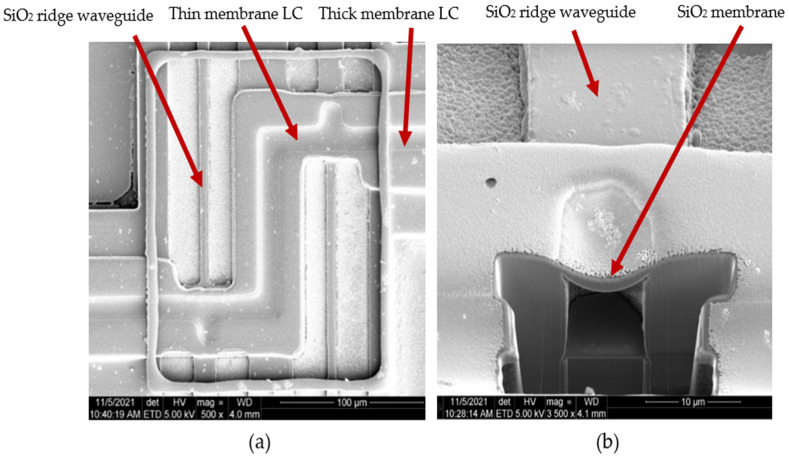
(**a**) Top view image of a scanning electron micrograph (SEM) of the thin membrane optofluidic region. (**b**) Profile of the protrusion cavity using focused ion beam milling (FIB). FIB backfilling accounts for increased thickness of the membrane in this image.

**Figure 6 micromachines-13-00721-f006:**
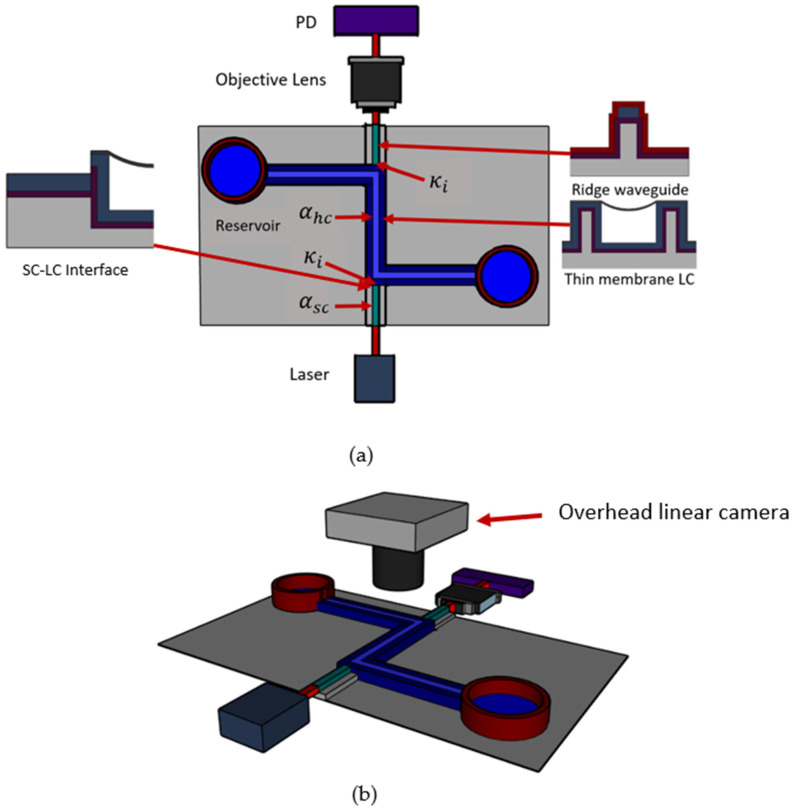
(**a**) Optical test bench setup for optofluidic waveguide characterization. Optical loss in the SC and LC channels were determined using 4 mm LC channels. $\alpha_{sc}$ is the loss in the solid core, $\alpha_{hc}$ is the loss in the hollow core, and $\kappa_{i}$ is the interface efficiency. (**b**) Loss test bench setup utilizing an overhead linear camera for top view scattering measurements.

**Figure 7 micromachines-13-00721-f007:**
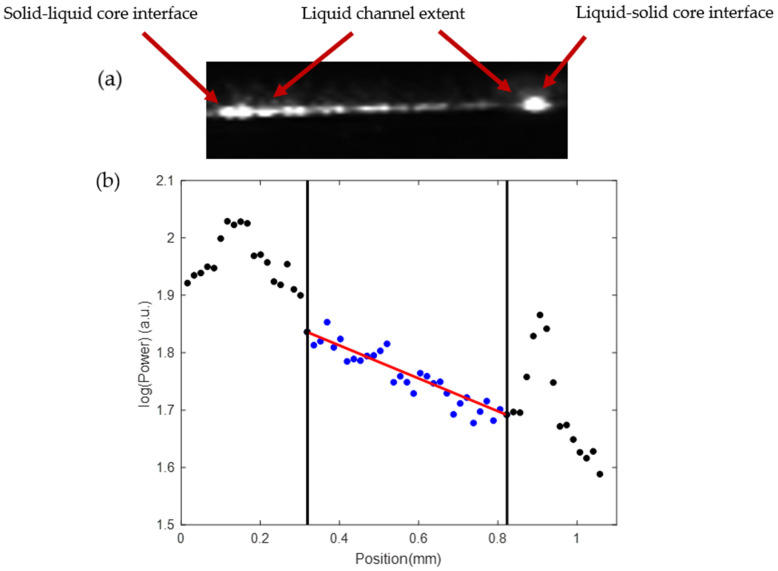
(**a**) Greyscale image of light propagating in the liquid channel. This image was used to determine the optical loss of the channel using top view scattering. (**b**) Intensity versus position plot using the top view scattering technique for a hollow core channel. The black points are solid to hollow waveguide interfaces. The interfaces cause scattering which results in significant loss. The blue points correspond to intensity values inside the LC region.

**Figure 8 micromachines-13-00721-f008:**
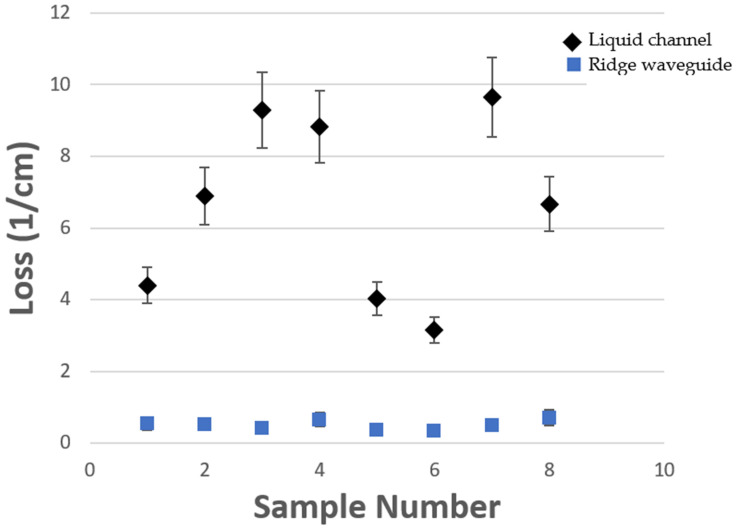
Liquid-channel (diamonds) and solid-core ridge waveguide (squares) loss coefficient values for eight devices with uncertainty bars.

**Figure 9 micromachines-13-00721-f009:**
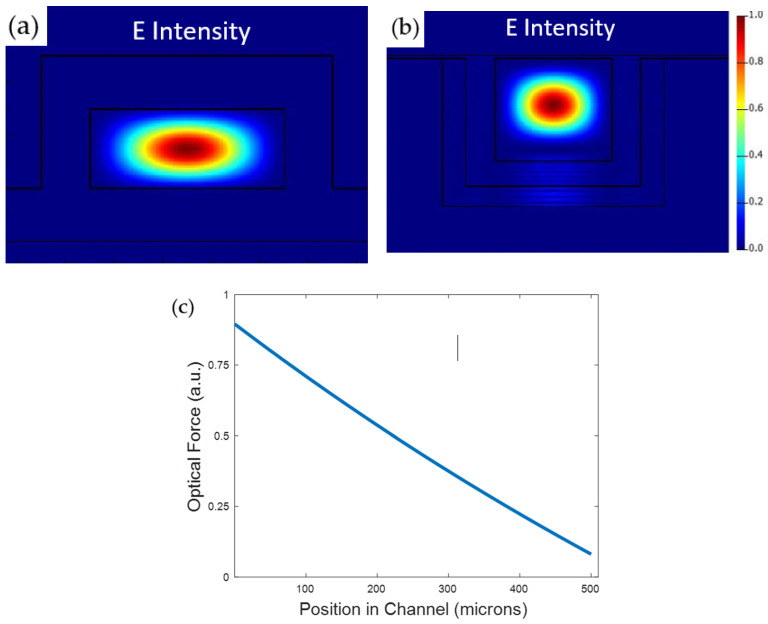
Simulated zero-order E field intensity mode profile for the SC (**a**) and LC (**b**) regions of the platform. (**c**) Simulated optical force spatial distribution. The optical force loss decrease is insignificant for proximity applications.

**Figure 10 micromachines-13-00721-f010:**
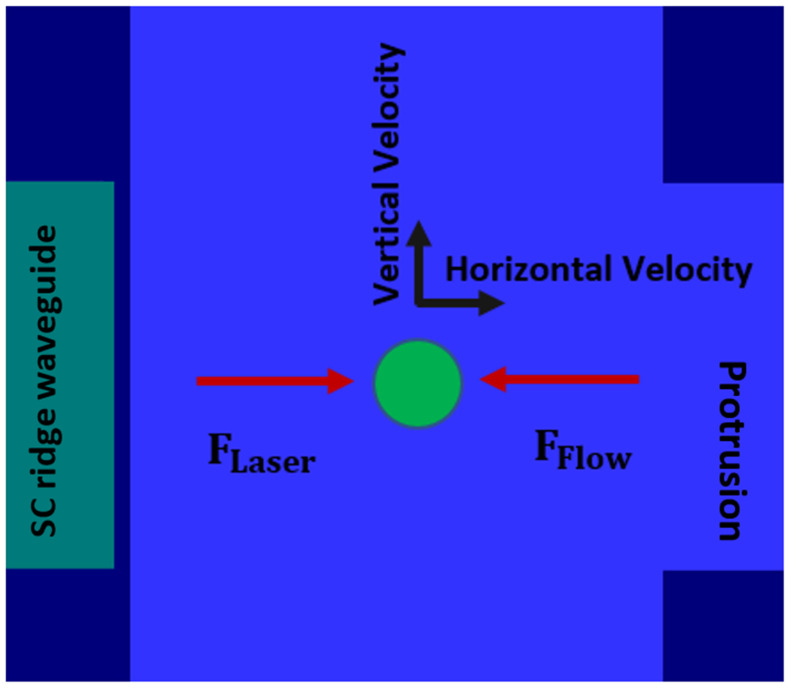
Forces acting on a particle in the optofluidic section of the device. Minimum laser force needed to trap a particle in the protrusion cavity.

**Figure 11 micromachines-13-00721-f011:**
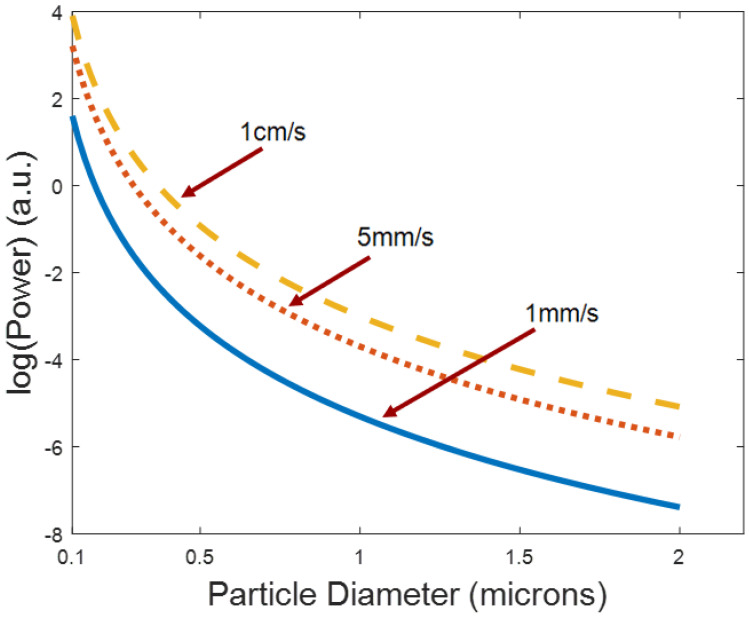
Minimum optical power for trapping of various sized particles. This is measured as power into the chip from the laser source. Typical SC waveguide lengths are between 3–4 mm with an average loss of 0.50 cm^−^^1^. Multiple flow velocities represented.

## Data Availability

Data is contained within the article.

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
