# Peer review of "Optofluidic Particle Manipulation Platform with Nanomembrane"

_micromachines, 2022, doi:10.3390/mi13050721_

Round 1

Reviewer 1 Report

The paper deals with the fabrication and preliminary characterization of an optofluidic particle manipulator on a silicon chip that features a 300 nm thick silicon dioxide membrane as part of a microfluidic channel. The device fabrication is novel but the manuscript does not present a characterization that exploits the full potentiality of the device. The paper is not so easy to read and figures are sometimes too simple and do not represent the complete picture of the optofluidic device.  Before publications major issues have to be carefully addressed:

Major issues:

1. The introduction is confusing and not well organized. It has to be re-written because it mixes properties of the presented device with literature results that seem uncorrelated. Authors should start with literature state-of-the-art, then present their device and then explain the advantages of their work.

2. The property of the encapsulated membrane are not demonstrated. Author have to add an experiment showing the advantage of this feature.

3. Simulations showing the waveguide properties of the SC, at the SC-LC interface and inside the LC have to be added.

Minor issues are:

1. Figure 8 is not clear to me: where is the profusion?

2. In the abstract authors claim that they can have sorting, but they do not show any experimental result on this topic.

3. A 3d- section of Figure 5 at the SC-LC interface would help the reader.

Reviewer 2 Report

In his paper, Zach Walker et al proposed a novel method to fabricate the thin membrane-formed microfluidic channel for the particle manipulation on the chip. The authors demonstrated detailed sections from fabrication to optical setup then to the result analysis. This holds great value for the future research in the field of analytical technology. To ensure the advantages, understandability and logic of this paper, a minor revision is required:

  1. What’s the advantages of thin-membrane microfluidic channel? The authors should give more introduction in the background.
  2. In Fig. 7, the loss of the liquid channel showed a significant variability, the author should analyse this.
  3. In Fig. 9, the author presented the relationship of particle diameter, flow rate and power. Is this the simulated results or the test data? If they are real tested data, what kind of particles are used? The authors need to make it clear.

Reviewer 3 Report

The paper presents an optofluidic platform that uses waveguides to exert radiation pressure to trap and manipulate colloidal particles. The topic is interesting. However, the presentation of the paper needs to be improved. The motivation of the work isn’t discussed very clearly. More importantly, the working principal of the device is not very clear. Potential applications of the work should also be suggested. In addition to these general issues, I have listed more specific comments below:

  1. The trapping mechanism is not very well discussed in the paper. For example, the equilibrium trapped position of a particle, and spatial distribution of the optical force profile has not been discussed. Please discuss this in detail. A numerical simulation of the optical force profile would greatly improve the quality of the paper.
  2. Silicon dioxide of two different refractive indices were used in the device. Could the authors please explain how they verified/measured the refractive indices?
  3. In the SEM figure (Fig. 4), please label the relevant structures and materials.
  4. The manuscript states that the overhead camera was used to capture images of the waveguides as light passes through them. Later image processing was used to determine power in each section of the waveguide. This is a very interesting approach. Please discuss this in detail. The images captured by the camera should be included in the manuscript. The algorithm used for image processing should also be discussed. If there are any prior papers in the literature that use similar approach, they should be cited.
  5. Figure 7 is missing legends. It is not obvious what the blue markers represent.
  6. Please mention the model number of the Thorlabs photodetector amplifier used for measuring the throughput.
  7. Please mention the model number of the laser source and its output power.
  8. How many different sized particles were tested to generate Figure 9? Without proper data markers, it is not possible to figure this out. The continuous nature of the plots suggest that a fitted curve was used. This should also be stated and explained. I suggest showing the actual data points with markers and then plotting the fitted curve on top.
  9. When measuring the trapping force for different flow rates, how are the authors ensuring that the particles get trapped at the same location for all trials? Since the optical power varies along the waveguide, it appears that the measured force would also depend on the location of the particle. Please discuss.
  10. Since the paper covers particle trapping and manipulation, some papers on near-field optical trapping and optoelectronic tweezers should be cited. For example:
    1. https://doi.org/10.1021/nn200590u
    2. https://doi.org/10.1039/C7SM01863K
    3. https://doi.org/10.1103/PhysRevA.100.013857
    4. https://doi.org/10.1038/nphoton.2011.98
    5. https://doi.org/10.1063/5.0020446

Round 2

Reviewer 1 Report

Authors have satisfied my previous my concerns except for the experiment showing the application of the nano-membrane developed in this work. So, please add an experiment using this nano-membrane.

Author Response

We believe we have answered all previous review concerns in our latest submission of the manuscript. The remaining concern of the review sent today is in regards to demonstrating an application of the thin membrane used in the construction of our device.  We have already added a section showing that a micropore can be drilled into the membrane.  We believe that further experiments to demonstrate the use of this micropore is beyond the scope of this paper. 

Reviewer 3 Report

The revised version addresses some of the issues I mentioned in my original review. A few small details still need to be corrected. The suggested minor revisions are:

  1. Please fix the formatting issues that exist throughout the revised paper. The figuresand captions appear incorrectly.
  2. The authors mention that some of the plots are simulated. It is not very clear how the experimental measurements were used to justify the results. Please clearly mention what data were used from the experimental measurement. Were any of the simulation parameters obtained from the experimental measurement? If so, please explain. If not, then clarify. 

Author Response

Response to Reviewers Comments

Point 1: Please fix the formatting issues that exist throughout the revised paper. The figure sand captions appear incorrectly.

Response 1: For some reason the formatting looks to be a lot better on the Word Document. When the file was converted to a PDF, the file seems to have changed all of my formatting, not sure why that is the case.

Point 2: The authors mention that some of the plots are simulated. It is not very clear how the experimental measurements were used to justify the results. Please clearly mention what data were used from the experimental measurement. Were any of the simulation parameters obtained from the experimental measurement? If so, please explain. If not, then clarify.

Response 2: Additional text was added to clarify which simulated plots used experimental data and which plots did not. See the 7th and last paragraph in section 4.
